# Recent Progress on Optical Biosensors Developed for Nucleic Acid Detection Related to Infectious Viral Diseases

**DOI:** 10.3390/mi14020295

**Published:** 2023-01-23

**Authors:** Ece Eksin, Arzum Erdem

**Affiliations:** 1Biomedical Device Technology Program, Vocational School of Health Services, Izmir Democracy University, 35290 Izmir, Turkey; 2Department of Analytical Chemistry, Faculty of Pharmacy, Ege University, 35100 Izmir, Turkey

**Keywords:** optical biosensor, nucleic acid biosensor, infectious viral diseases, surface plasmon resonance, colorimetry, fluorescence

## Abstract

Optical biosensors have many advantages over traditional analytical methods. They enable the identification of several biological and chemical compounds directly, instantly, and without the need of labels. Their benefits include excellent specificity, sensitivity, compact size, and low cost. In this review, the main focus is placed on the nucleic acid-based optical biosensor technologies, including colorimetric, fluorescence, surface plasmon resonance (SPR), Evanescent-Wave Optical, Fiber optic and bioluminescent optical fibre. The fundamentals of each type of biosensor are briefly explained, and particular emphasis has been placed on the achievements which have been gained in the last decade on the field of diagnosis of infectious viral diseases. Concluding remarks concerning the perspectives of further developments are discussed.

## 1. Introduction

Human health is seriously threatened by infectious diseases in both developed and developing nations. For instance, COVID-19 has caused an enormous influx in virus research, especially for advanced testing and early diagnostics. The X-ray computed tomography (X-ray CT), single photon emission computed tomography (SPECT), and positron emission tomography (PET) are current techniques for virus imaging which are costly and have low resolution. Although enzyme-linked immunosorbent assay (ELISA) or reverse-transcription polymerase chain reaction (RT-PCR) are the most common techniques, they involve several time consuming and costly steps, such as nucleic acid amplification and purification, and require trained personnel. In a study by Maddali et al. [1], a comparison of various molecular imaging methods is covered in detail.

DNA is an appealing biomolecule for many applications because of its unique double-helical structure, from which reversible complementary sequences can be identified. Numerous applications of DNA have been discovered since the genomics revolution, including DNA nanowires and nanomotors [2,3]. In particular, a great number of biosensors have been researched and developed above all other DNA-based applications [4,5]. Nucleic acid-based biosensors and biochips which are micro-scale or nano-scale have recently gained attention in biotechnology. In basic research about the genetics of diseases as well as in more practical applications as medical diagnosing, the miniaturized biosensor and biochips has become very popular and preferred tools. In comparison to conventional bioassays, nucleic acid-based biosensors and biochips provide various advantages to genetic testing, including quick analysis times, low liquid handling, smaller sample volume requirements, and multiple detection options for specific nucleotide sequence of DNA or RNA (i.e., cDNA, miRNA, mRNA, tRNA), circulating tumor DNA, single nucleotide polymorphisms (SNPs), damaged DNA, etc. [6,7,8]. Nucleic acid-based biosensors serve as effective screening tools for determination of a trace low-molecular-mass analytes (i.e., drugs, toxins, and pesticides) as an important aspect of studies in drug discovery and environmental processes [9]. On the other hand, protein binding activity can be processed, and transformed into a measurable signal using nucleic acid-based biosensors that can recognize and detect a specific targeted protein [10,11,12].

Among several types of biosensors, optical biosensors exhibit good performance in detecting biological systems and promote significant advances in clinical diagnostics due to the advantages of optical signal, such as high sensitivity, immunity to external disturbance, stability, and low noise [13]. These biosensors offer additional advantages including reliability, durability, and the possibility to be combined onto one chip without complexity in the pretreatment and likely effect on the nature of the target molecules. Due to its safe, easy-to-use, and cost-effective technology, optical biosensors offer an alternative technique for viral detection. They even eliminate the requirement for nucleic acid amplification. HIV, Ebola, norovirus, influenza virus, and other pathogens have all been detected by surface plasmons, colorimetry, and fluorescence-based methods.

Sensitivity and detection limit (DL) are crucial factors in defining a biosensor’s performance in the sensor development. The magnitude change in the transduction signal in response to the change in analyte is known as sensitivity, which is referred to as the strength of the light–matter interaction. On the other hand, the resolution of a read-out system determines the DL, which is the lowest detectable quantity of analyte. The specification of the DL for optical biosensors is discussed in detail by Chen and Wang [13]. Optical biosensors may easily miniaturized and be used as point-of-care (POC) diagnostic devices. POC diagnostic tools allow to use samples without any preliminary sample treatment, and do not require trained staff or expensive equipment.

Enhancing optical imaging has the potential to be utilized to detect and monitor viruses to develop treatment options more quickly and efficiently. The use of these imaging techniques for the detection of infectious viruses has already begun. However, further investigations are required to bring these technologies from research laboratories to market. This review focused to different optical techniques and presented a perspective to their potential use for optical detection of infectious viruses.

## 2. Infectious Viral Diseases

Viruses are obligatory intracellular parasitic subcellular infectious organisms. To replicate themselves, they invade and control a host cell. The information that a virus needs to replicate and produce new virions after being introduced into a vulnerable cell, is included in its nucleic acid genome. The information that a virus needs to replicate and generate new virions after being introduced into a susceptible cell is found in its nucleic acid genome [14].

It is possible to identify three major groups of viruses, each of which may have a separate evolutionary history. Poxviruses and herpesviruses belong to a class of viruses that have DNA as their genome, whether it is single-stranded or double-stranded. The DNA genome is replicated by direct DNA to DNA copying. The second class of viruses contains RNA as their genome, and direct RNA to RNA copying is used to duplicate the RNA. All RNA viruses require virus-encoded proteins to build a replicase that replicates the viral RNA, since cells do not contain RNA to RNA copying enzymes. The third class of viruses which also have an RNA–DNA stage in their life cycle, encodes the enzyme reverse transcriptase [14].

Viruses are able to spread from one infected organism to another in a variety of ways, including oral–fecal, airborne, bloodborne, sexual, and congenital. When an infected individual sneezes or coughs, the virus is aerosolized and spread through the air as small respiratory droplets or dust particles. In case of contaminated air is inhaled or when virus-containing mucosal secretions is transferred to mucosal surfaces, infection may begin. Some people carry viruses in their blood that are known as blood-borne viruses (BBVs) and can be transmitted from one person to another. BBVs are primarily spread sexually or via direct contact with infected blood or other bodily fluids. Sexual contact is a common way for viruses to spread. A virus may be found in genital area, or in semen or vaginal secretions. The vaginal mucosa is where the infection starts; however, it can spread to other organs. In case of congenital infection, viral infections, including those caused by human immunodeficiency virus (HIV), cytomegalovirus, and rubella virus, can arise during the birth process or while the fetus is in utero. Breast-feeding can also cause vertical transmission soon after birth [14,15].

Virus-based infectious illnesses have the potential to spread to pandemic levels that threaten both the global population and daily life. To limit the loss of life globally, it is crucial to early diagnosis, treatment, and prevention of spreading the pandemic infectious diseases not further the community.

### 2.1. Severe Acute Respiratory Syndrome (SARS)

In March 2003, the World Health Organization (WHO) determined the severe acute respiratory syndrome (SARS) outbreak in China’s Guangdong province as a threat to the whole world. Over the following months, SARS spread to more than 30 countries, become the first pandemic of the twenty-first century. SARS was unique among novel pathogens, including H5N1 influenza, Hantavirus, Nipah virus, and Avian flu, as it had the ability for efficient person-to-person transmission [16]. A broad class of viruses known as corona viruses frequently infects people. In December 2019, the first severe acute respiratory syndrome coronavirus 2 (SARS-CoV-2) patient was reported. On 11 March 2020, the WHO declared the COVID-19 outbreak to be a pandemic disease [17,18].

### 2.2. Influenza

The influenza A or B virus is the primary cause of influenza, an acute respiratory disease. The entire world is impacted by a pandemic, epidemic, or seasonal pattern, which causes enormous morbidity and mortality [19]. A total of 864 cases and 456 deaths of influenza A (H5N1) human infection have been reported worldwide from 18 countries since 2003 to 31 March 2022 by WHO.

A particular strain of the influenza A virus is called swine flu, found in pigs. Influenza A (H1N1), a strain of flu virus, which caused respiratory infections in humans in 2009 and spread quickly throughout the world. In August 2010, the WHO declared the swine flu pandemic to be over. H1N1 virus was also responsible for the 1918 flu pandemic.

Influenza B viruses cause around one-fourth of clinical infection cases each year. During flu seasons, influenza B viruses are occasionally the predominant circulating strains, as was the case in the 2019–2020 United States flu season, influenza B caused more than 50% of the cases [20].

In contrast to previous diagnostic modalities, rapid influenza diagnostic tests quickly identify influenza viral antigens and screen individuals with suspected influenza. The method that is used the most frequently is the one that relies on immunologic methods to identify viral antigens in patients’ respiratory secretions. All rapid tests are simple and can deliver results in 30 min. The ability of each test to distinguish between influenza A and B varies. However, until now, these tests have been unable to distinguish between different influenza A strains, such as H1N1 and H3N2 [19].

### 2.3. Middle East Respiratory Syndrome (MERS)

The Middle East Respiratory Syndrome Coronavirus (MERS-CoV), which causes MERS, was first discovered in Saudi Arabia in 2012. According to the WHO, since 2012, 27 countries have reported 858 known deaths due to the infection and related complications [21]. Regardless of clinical signs and symptoms, the diagnosis is based on test confirmation of MERS-CoV infection. Each year, two million people (including one million pilgrims) visit Mecca and Medina from 180 different countries. To detect the real spread, surveillance must be carried out rapidly, precisely, and consistently. Widespread use of diagnostic tests and surveillance is crucial if the right precautions are to be followed [22,23].

### 2.4. Hepatitis

Hepatitis is characterized as liver inflammation that can be caused by several factors, including excessive alcohol use, autoimmune conditions, medicines, or pollutants. However, viral hepatitis is the most common cause of hepatitis and results from a viral infection [24]. Hepatitis A, B, and C are the three viral hepatitis strains that are most prevalent in the United States. Hepatitis D, E, and G are the other forms of viral hepatitis; however, they are less common. Hepatitis A, C, D, E, and G are RNA viruses from different families, whereas the hepatitis B virus is a DNA virus. Chronic hepatitis C infection is a cause of mortality that contributes to over 20,000 fatalities per year [25]. Seventy million individuals worldwide have active viraemia, and the prevalence of HCV is between 2% and 3% [26]. Due to inadequate control management, the prevalence of HCV infection is considerable in developing countries. Early detection is the first step in the fight against HCV.

### 2.5. Human Immunodeficiency Virus (HIV)

Acquired immunodeficiency syndrome (AIDS), which attacks the immune system of the body, is caused by the human immunodeficiency virus (HIV). AIDS was first recognized as a new disease in 1981. Ever since, the reasons for its unexpected emergence, epidemic spread, and special pathogenicity have all been the focus of extensive research [27]. According to the WHO’s 2022 report, around 38.4 million people were reported to live with HIV/AIDS worldwide and 650,000 people died from HIV-related causes in 2021 [28].

In a patient’s blood or plasma, HIV can be diagnosed by detection of antibodies that indicate the presence of viral nucleic acid, either by PCR chain reaction, p24 antigen, or growing viruses in cell culture. The most frequent method of determining HIV infection is an antibody test [29].

### 2.6. Human Papilloma Virus (HPV)

Sexually transmitted infections (STIs), which affect 50% to 70% of sexually active people globally, have a high rate of morbidity and death. The most prevalent viral sexually transmitted infection is considered to be human papillomavirus (HPV) infection globally. Additionally, more than 5% of cancers worldwide are linked to persistent HPV infection [30]. Generally, HPVs result in acute, non-persistent infections. Therefore, just like other oncoviruses, there is a considerable difference between the time of the chronic infection’s detection and its early stages. HPV infections are identified by demonstration of HPV genomic sequences in infected tissues or by locating HPV proteins in tissues. Since HPVs cannot be grown or isolated in tissue culture, one of the most used methods for detecting viruses cannot be used to detection of HPVs [31]. Most often, DNA amplification-based approaches are used to identify HPV genomic sequences. In these methods, specific viral sequences are first amplified by PCR, before identification. The analytical sensitivity, clinical applicability, complexity, and trustworthiness are the limitations of the methods.

### 2.7. Dengue

In tropical and subtropical climates, the arboviral infection dengue, which commonly transmitted by the mosquito *Aedes aegypti*, is hyperendemic. Approximately half of the world’s population is at risk due to recent increases in its incidence. The diagnosis can be performed either directly by screening viral components in the blood or indirectly by serological tests. The early stages of a viral infection require very sensitive viral component identification. Viral nucleic acid in serum or the virus-expressed soluble non-structural protein 1 (NS1) can be detected by RT-PCR and ELISA, respectively [32].

### 2.8. Ebola

Ebola virus (EBOV) is the causative agent of EBOV disease (EVD) with a case mortality rate as high as 89%. Due to the large population of migrants, the illness has become a global public health threat. The EVD symptoms are not distinctive from those of other endemic diseases [33,34]. The patients first exhibit mild influenza-like symptoms, followed by shock and multiorgan failure. There is an urgent need for sensitive, quick diagnoses for EVD due to the high mortality and extensive symptom overlap.

## 3. Diagnostics of Infectious Viral Diseases with Optical Biosensors

A device that combines a nucleic acid as the biorecognition component and a transducer for converting the biorecognition event into a measurable signal is known as a nucleic acid-based biosensor. These biosensors have recently become more popular due to their extensive use in biomarker monitoring, which is crucial in the area of clinical diagnostics. DNA, RNA, peptide nucleic acid (PNA), zip nucleic acid (ZNA), and aptamers are the several types of nucleic acid molecules. The basic principle of nucleic acid biosensors is based on sequence complementarity according to Chargaff’s base pairing laws, which corresponds to the hybridization process [35,36]. On the other hand, aptamers operate more like antigen–antibody or receptor–ligand interactions in terms of their detection principle, which is frequently accompanied by conformational changes in the aptamer.

In order to develop a nucleic acid-based biosensor, nucleic acids are immobilized onto a solid substrate by adsorption, covalent bonding, or ionic interaction, then connected to a physico-chemical transducer. Immobilization can be performed in a variety of ways, including adsorption to inert carriers, chemical cross-linking to particles (i.e., magnetic beads), physical entrapment in a gel-like lattice, and microencapsulation in nanospheres. Probe direction and easy access to the target molecule are two important events of immobilization [37,38,39]. In the case of solid-state hybridization, different fabrication techniques, such as adsorption, covalent bonding, and avidin/streptavidin-biotin interaction on the electrode surface, are available to attach DNA probe to transducer surface [40]. However, numerous factors (such as ionic strength, temperature, solubility, and catalyst concentration) must be considered for solution-based hybridization assays and must be optimized subsequently.

DNA generates biological recognition layers that are very durable and reusable, unlike other sensing molecules such as enzymes or antibodies, which are difficult to synthesize. The effectiveness of DNA-based biosensors can be increased by a clever and sophisticated nucleic acid recognition procedure using aptamers and peptide nucleic acids (PNA). PNA is a synthetic nucleic acid analog with an N-(2-aminoethyl)glycine backbone connected by peptide bonds. PNA/DNA and PNA/RNA hybrid structures are more stable than the comparable DNA/DNA and DNA/RNA hybrids due to their uncharged backbone. Additionally, PNA exhibits chemical and thermal stability, resistance to ionic strength, pH changes and intracellular enzymatic cleavage [41].

PNA probes are a promising alternative for DNA and RNA probes because they are strong, and effectively hybridize with the complementary DNA strands.

Zip nucleic acids (ZNA) are oligonucleotide-oligocation conjugates in which the nucleic acid oligomer has multiple cationic spermine moieties attached. ZNAs are able to distinguish between a perfect match and a complementary sequence with one base pair mismatched [42,43,44]. A new generation nucleic acid: ZNA was applied for the first time by Erdem and Eksin in 2019 [45] for electrochemical monitoring of nucleic acid hybridization related to Factor V Leiden mutation based on magnetic beads assay. The application of ZNA was then shown in different nucleic acid-based biosensor designs [46,47,48].

Aptamers can be chemically synthesized, modified, and combined with a variety of nanomaterials easily. DNA aptamers attach to a variety of bioanalytes, including nucleic acids, cells, proteins, viruses, and small molecules, with excellent selectivity and affinity. Additionally, it has been proved that they are able to distinguish enantiomers. However, due to the 2’ hydroxyl functional group in RNA aptamers, they offer higher binding affinity than DNA aptamers to the identical target sequence. Aptamers have high levels of reproducibility and inexpensive manufacturing costs, can tolerate extreme environmental conditions, and be kept without extra precautions. An in vitro technique known as the systematic evolution of ligands by exponential enrichment (SELEX) is used to choose potential aptamer candidates. The goal of SELEX is to produce specific oligonucleotides that bind to the target [49]. In case of aptamer-based biosensors, the secondary and tertiary structures of the aptamer undergo a conformational change as a result of the biorecognition event between aptamer and target molecule. These structural changes are responsible for generating a signal for display. 

Figure 1 depicts an overview of optical sensors for the diagnosis of viral infectious diseases.

### 3.1. Colorimetric Biosensors

Colorimetric biosensors may identify the presence of certain substances using an optical detector or a color shift that is visible to the naked eye. Colorimetric biosensors are excellent candidates for POC diagnostics because of their simplicity of use without costly analytical apparatus [50,51,52].

Gold nanoparticles (AuNPs) are frequently used for fabrication of colorimetric biosensors using aggregation (or deaggregation) of the nanoparticles induced by formation of covalent or non-covalent bonds with the target substance. The aggregation of gold nanoparticles will change the color of the solution from wine red to blue, which corresponds with surface plasmon band changes from 523 nm towards 610–670 nm [50]. Therefore, the change at color from red to blue can be obtained as a convenient signal readout using AuNPs. The Frens and Turkevich method [53] makes it simple to create spherical AuNPs that are stable, consistent in size, and well-dispersed. Therefore, AuNPs-based analysis is inexpensive and easy-to-use. The crosslinking colorimetry of DNA-AuNPs (CCDA) and the salt-induced colorimetry of unmodified AuNPs (SICUA) are the two basic types of AuNPs-based nucleic acid colorimetric detection techniques [54,55]. DNA-AuNPs probes can be self-assembled into a networked DNA-nanoparticle system by a complementary DNA duplex with “sticky ends” as a linker strand. This method enables the assembly control of DNA-AuNPs with a reversed color by thermal denaturation. On the other hand, SICUA uses unmodified AuNPs, in order to avoid covalent functionalization of AuNPs and simplify the experimental stages. The key mechanism of the SICUA is the electrostatic interaction between unmodified AuNPs and nucleic acids. Both the CCDA and SICUA methods have benefits and drawbacks. For instance, SICUA can provide a color signal that is more visible, but it requires precise single and double DNA variation both before and after the nucleic acid hybridization event. CCDA has a wider range of applications, but it requires long time and more effort for preparation of DNA-AuNPs [56].

Unmodified AuNP was used for detection of dengue virus based on peptide nucleic acid (PNA)/DNA hybridization by Rahman et al. [57]. In the absence of target DNA, due to the neutral charge of PNA which is on the surface of AuNPs, unmodified AuNPs immediately aggregate and show purple color. After hybridization of PNA and DNA target, negatively charged hybrid complex occurred and dispersed the AuNPs to give red color. The detection limit for dengue virus DNA was reported as 0.12 μM. In many reports, AuNPs have been functionalized to take advantage of the color change by the surface plasmon resonance effect phenomenon [58]. For instance, Teengam et al. reported the multiplex colorimetric paper-based analytical device (PAD) for DNA detection based on the aggregation of AgNPs induced by D-proline/2-aminocyclo-pentanecarboxylic acid (known as acpcPNA). The negatively charged AgNPs interacted with the cationic PNA probe, causing the AgNPs to aggregate and change in color. MERS-CoV, MTB, and HPV were all simultaneously detected using the suggested sensor by Teengam et al. [59].

Loop-mediated isothermal amplification (LAMP) method which was developed for DNA detection by Notomi in 2000 [60] offers rapid detection, high efficiency, cost effectiveness, and reproducible results. SYBR Green I, hydroxynaphthalene blue (HNB), or propidium iodide can be used to see color changes with the naked eye. Color changes in case of positive reactions from orange to green, purple to blue, orange to pink in the presence of SYBR Green I, HNB and propidium iodide, respectively. A simple reverse transcription-LAMP (RT-LAMP) HNB dye-based visual detection assay of pandemic influenza A H1N1 virus infection described by Ma et al. [61]. In the study of Kumvongpin et al. [62] gold nanoparticles (AuNPs) used in conjunction with LAMP for the detection of high-risk human papillomavirus (HR-HPV) genotypes; HPV16 and HPV18. Visual turbidity was compared with AuNP color change for the detection of the LAMP product. For both HPV genotypes, the sensitivity of the LAMP-AuNP assay was up to 10-fold higher than that of the LAMP turbidity assay. The entire assay time for visual detection reported as 25 min.

Alkaline phosphatase (ALP) detected using a colorimetric test approach based on enzyme-induced metallization by Zhou et al. [63]. This method combined with immunomagnetic separation and further used for sensitive detection of avian influenza virus. The main principle of the developed technique was that the enzyme could induce a silver shell to be deposited onto the surface of AuNPs, which could cause a significant color change. This phenomenon helped the bio-metallization process in which ALP catalyzed silver deposition on the surface of AuNP seeds, and AgNPs exhibit greater extinction coefficient than AuNPs. When a longer incubation period or higher ALP concentration was used, the detection solution’s color would change to brown or even black, which contributed to the further growth of the Ag shell.

An overview of studies regarding the effectiveness of nanoparticles for detection of coronaviruses was presented by Nikaeen et al. [64]. They reported that gold, silver, silver sulfide, titanium oxide, zirconium, graphene, and some biopolymeric compounds have been the most applicable nanomaterials for this goal [64]. For instance, a colorimetric nanoparticle-based assay developed by Kumar et al. [65] to detect the presence of the RdRp gene of SARS-CoV-2. The formation of the oligo probe-target complex within samples with SARS-CoV-2 pharyngeal RNA samples can be visualized using AuNPs, resulting in a pink-to-blue color change. The detection limit was 0.5 ng of SARS-CoV-2 RNA and results can be obtained within 30 min [65].

Due to their unique structure and characteristics, such as superior conductivity, high mechanical stability, and ease of functionalization, two-dimensional (2D) materials have been widely used for the development of different colorimetric sensors and biosensors. Hemin-modified graphene nanosheets were synthesized by Guo et al. [66] via a wet chemical process. This hybrid material used as a practical, and eco-friendly colorimetric platform for label-free detection of the Hepatitis B virus (HBV). Recent research progress on the design, fabrication, and applications of 2D material-based colorimetric biosensors were reviewed by Zhu et al. [67] in detail.

Table 1 presents some of the colorimetric biosensors that have been developed for detection of different viruses using different recognition elements, along with their dynamic detection range and detection limits [57,59,62,68,69,70,71,72,73,74,75,76].

### 3.2. Fluorescence-Based Biosensors

Fluorescence-based sensing and imaging has special benefits that make it applicable in research and clinical applications, such as high temporal resolution, strong sensitivity, and the availability of biocompatible imaging agents. Fluorescence-based optical biosensors comprise the broadest type of sensors due to the commercial availability of numerous fluorescent probes, optical fibers, and appropriate optical apparatus. Additionally, fluorescence is an inexpensive, accessible tool since a high excitation power is not necessary [77,78,79,80,81]. Most analytical signals are produced from real-time observations of fluorescence intensity, which describe the usual behavior of many molecules over time. Fluorescent probes including fluorophores, quantum dots (QDs), nanoclusters, carbon dots, and fluorescent nanomaterials, show either enhanced or quenched fluorescence intensity in respect to the target [82].

The conformational and/or structural alterations of aptamers can have an impact on a dye’s fluorescence or the fluorescence resonance energy transfer (FRET) between two dyes through direct modification with fluorophores. Quantitative measurement of the target concentration can be accomplished by the signal change, which can be either an increase (signal-on) or a decrease (signal-off). A fluorescent aptasensor developed by Pang et al. [83] for the sensitive detection of the recombinant hemagglutinin (rHA) protein of the H5N1 influenza virus in human serum. In a complex matrix or in the presence of other proteins, the system demonstrated good selectivity and sensitivity by means of the combination of the high selectivity of the rHA aptamer with the high fluorescence amplification capacity of the core–shell Ag@SiO_2_ NPs.

Researchers investigated signal amplification techniques during the creation of fluorescent biosensors, which dramatically improved the fluorescence signal and sensitivity of the biosensors. This approach allowed for the correct analysis of trace biomolecules in biological samples. Cyclic signal amplification (CSA) technology is one of the most practical signal amplification techniques due to its basic operation and inexpensive cost. For very sensitive and low limit of detection for biomolecules, the fluorescence signal might be amplified several times following the CSA technology [84]. The most widely used CSA techniques include enzyme-assisted amplification (EAA), strand displacement reactions (SDR), and rolling circle amplification (RCA) [85,86,87]. Examples include the simultaneous detection of enterovirus 71 (EV71) and Coxsackievirus B3 (CVB3) by Du et al. [88] using the RCA method, with low detection limits of 4.1 × 10^3^ and 1.9 × 10^4^ copies/mL, respectively. A method for identification of three viruses (Ebola, Zika, and Dengue) using the RCA technique was developed by Ciftci et al. [89] and it showed a considerable influence on the diagnosis of associated diseases. Wen et al. [90] developed a sensitive and selective fluorescent assay based on RCA method for detection of Ebola virus protein VP40. Using the developed assay, LOD was found as 1.4 pM with a linear range from 30 fM to 3 nM.

Lu et al. [68] designed a gold nanorods (AuNRs)-based fluorescent biosensor for the detection of hepatitis B virus DNA sequences. The addition of fluorescein (FAM) -tagged single-stranded DNA (FAM-ssDNA) in AuNRs suspension caused FAM-ssDNA-CTAB-AuNRs ternary complex formation. A fluorescence resonance energy transfer (FRET) process from FAM to AuNRs occurred by resulting structure and the fluorescence intensity of FAM was then quenched.

Carbon dots (CDs) were employed as fluorescent NPs quenched with AuNPs in Qaddare and Salimi’s FRET-based HIV biosensor [91]. This biosensor exhibited a dynamic range of 50.0 fM to 1.0 nM and LOD was 15 fM. Table 2 presents some of the fluorescent-based biosensors that have been developed for detection of different viruses using different recognition elements, along with their dynamic detection range and detection limits [68,83,90,91,92,93,94,95,96,97,98].

### 3.3. Surface-Enhanced Raman Spectroscopy-Based Biosensors

As an optical bioimaging approach, enhanced Raman scattering efficiency has significant potential for enabling deep and high-resolution volumetric imaging of biological tissues. Surface-Enhanced Raman Spectroscopy (SERS) has been used to detect a variety of viruses, including the influenza, Adeno, West Nile, and Rift Valley fever viruses. It offers better sensitivity and chemical specificity than the other optical detection techniques [99,100]. There are two categories of SERS-based techniques: direct and indirect. Direct or label-free approaches, which do not use reporter molecules, rely on identifying the spectrum of the analyte. However, due to the various and unexpected enhancement of components, direct sensing in biofluids can provide spectra that is difficult to understand [101].

### 3.4. Surface Plasmon Resonance-Based Biosensors and Localized Surface Plasmon (LSP)-Based Biosensors

The Surface Plasmon Resonance (SPR) technology has been in use for four decades. In 1983, SPR was initially employed to build an SPR-based sensor to detect biomolecular interactions [102]. Multiple manufacturers currently produce SPR instruments, and the SPR-based biosensor is the main optical biosensing technique. The SPR phenomenon occurs on the surface of metal (or other conducting materials) at the interface of two media (typically glass and liquid) in case of illumination by polarized light at a precise angle. This produces surface plasmons, which reduce the intensity of reflected light at a specific angle known as the resonance angle. This effect is proportional to the mass on the surface. A sensorgram can be created by plotting the change in reflectance, angle, or wavelengths versus time. The SPR technique provides direct, label-free, and real-time changes in refractive index at the sensor surface that is proportional to biomolecule concentration in all configurations. A practical SPR instrument consists of an optical detector that measures intensity shift, a sensor chip with a gold surface and a layer enabling ligand immobilization, and a fluidics system that allows for flow-through operation [102].

Surface plasmons are collective oscillations of the electron cloud at the surface of a metal excited by incident electromagnetic radiation. Different kinds of surface plasmons are associated with various metal structure. For instance, metal films support propagating surface plasmon polaritons (SPPs), whereas metal NPs promote localized surface plasmon resonances (SPRs or LSPRs) [1].

Localized SPR (LSPR) is based on metallic nanostructures (MNPs), such as Au and Ag, which have distinct optical characteristics that are not present in larger metal structures. The red color of aqueous dispersions of colloidal gold particles, which is an expression of LSPR, is a particularly remarkable example of such a phenomenon [102]. The loop-mediated isothermal amplification (LAMP) approach is more advantageous among colorimetric viral detection techniques due to its fast detection, high efficiency, lower cost, and greater dependability with highly repeatable test findings. Hollow Au spike-like NPs were used as the LSPR agent and multifunctional DNA served as the recognition element in H5N1 biosensor that developed by Lee et al. [103]. The three-way junction functions of the DNA employed in the study which were NP binding, recognition element, and signal enhancement. In another work, a method for COVID-19 clinical diagnosis that integrates LSPR with the plasmonic photothermal effect was reported by Qiu et al. [104]. The technique involved immobilizing complementary DNA for SARS-CoV-2 onto 2D-Au nanoislands (AuNI) to enable sensitive and specific detection of viral sequences via nucleic acid hybridization. With a low detection limit of 0.22 pM, the sensor demonstrated accurate virus detection.

New materials-based SPR sensors are currently being developed. For instance, various metals are employed to enhance particle formation and optimize plasmonic performance, such as gold film or particle-based SPR platforms. For instance, SPR platforms based on graphene have outstanding analyte adsorption, superior corrosion resistance, and high temperature resistance. According to reports, aptamer-based SPR sensors have a remarkably high analytical specificity. Compared to their 2D competitors, three-dimensional (3D) structure-based SPR platforms have a larger surface area, which increases the possibility of the analyte binding to the platform. As a result, SPR biosensors are frequently utilized as an initial diagnostic tool since they provide effective and high-throughput diagnostics because to their quick detection time and lack of labeling requirements [105]. Kim et al. suggested an SPR-based biosensor for the detection of AIV utilizing a DNA hybridization procedure. The study’s methodology involved measuring the amount of thiol linked oligonucleotides during the hybridization process [106].

This sandwich aptamer approach was recently used by Shin et al. [107] to design an SPR sensor for virus detection. An aptamer was immobilized to a paper strip sensor as part of the SPR platform. When the analyte contacted the sensor, sensor became red, and the SPR principle allowed one to quantify the color shift. The limit of detection was 10^3^–10^4^ CFU/mL.

Table 3 summarize some of the developed Surface-Enhanced Raman Spectroscopy (SERS), Surface Plasmon Resonance (SPR)-based biosensors for the detection of various viruses using various recognition elements with their dynamic detection range and detection limits [108,109,110,111,112,113,114,115].

### 3.5. Other Optical Biosensors (Evanescent-Wave Optical Biosensors, Fiber Optic Biosensors, Bioluminescent Optical Fibre Biosensors)

Fiber Optic Biosensors (FOBS) detect the target (bio)molecule by using optical transduction techniques using optical fibers as the transduction component. The notion of total internal reflection (TIR) is employed in fiber optics to correlate the light intensity measured at the detector with the initial target concentration [116]. Fiber-optic biosensors are based on the measurement of absorbance, RI, fluorescence, chemiluminescence, etc., and can be used for different biomolecule detection. With recent discoveries in the field of nanotechnology, the development of optical fibers with micro- or nano-sized dimensions has opened up a new route for detecting microorganisms, cells, in vivo biosensing [117].

In evanescent wave biosensors, specific ligands are immobilized to the sensor surface, and a solution of receptor is injected over the top. Subsequently, the binding is measured by recording changes in the refractive index, caused by the molecules interacting near the sensor surface within the evanescent field [118]. Evanescent wave fluorescence biosensors have evolved into a wide variety of devices since their discovery and initial application in the middle of the 1970s. Taitt et al. provided a comprehensive study of the evanescent wave fluorescence biosensors’ characteristics, operation, and applications [119]. Evanescent wave FOBS rely on evanescent wave detection methods. Total internal reflection at the exposed surface is how electromagnetic waves move within an optical fiber. The guided field in the core and the exponentially decreasing evanescent field in the cladding constitute the two components of light that propagate through an optical fiber. The cladding of a fiber is diminished or removed in evanescent wave FOBS, allowing the evanescent wave to interact with the surroundings. Thus, evanescent wave FOBS can rapidly and accurately detect these target analytes from complicated matrix samples, significantly improving the sensitivity, selectivity, and speed of the detection process.

Luminescence from fluorescent proteins or luminescent enzymes is commonly used in biological research for monitoring changes in the surroundings of cells. A simple method is to directly measure the luminescence from the sample solution in a test tube using a single photon detector by simple coupling optics. The luminescence detection with the optical-fiber-based system enables to fully use the merits of compactness, high quantum efficiency, and low noise of avalanche photo diodes detectors [120]. Recombinant bioluminescent cells are used in the bioluminescent optical fiber technology, and an optical fiber is used to transmit the bioluminescent signal from the analyte. The detection of a particular molecule may then be performed by accurately detecting the emitted photons with a photon detector. In order to obtain a high sensitivity to the specific molecules or ions, a highly effective luminescence detection is required. The capability of detection in a low concentration of sample solution can be upgraded by an increase in sensitivity.

## 4. Conclusions and Future Prospects

In the past decade, sensor design and biospecific coatings have significantly advanced in optical biosensor technology. Many applications have been reported and here their application for infectious viral disease diagnosis has been discussed.

In terms of identification specificity, oligonucleotide-based biosensors exhibit a lot of potential. Aptamers are useful for the detection of intact virus particles, whereas antisense oligonucleotides are capable of precisely detecting the pathogen strain at the genome level. In comparison to proteins, oligonucleotides are more advanced in chemical synthesis with a wide range of modifications.

The optical detection principle enables the construction of sensitive, straightforward, and affordable analytical devices with a wide variety of potential applications in portable biosensor systems for in situ screening and monitoring. This presents the biggest obstacle for further research and advancement in optical biosensor field. Future predictions for novel analytical approaches include multiplex assays that integrate the detection of several biomarkers or secondary disease biomarkers emerging from the progression of infection. The configuration described above could be used as a double confirmation test or to check the patient’s health.

A considerable development has been achieved, which is proved with the increasing number of publications addressing novel or improved sensor configurations. Although some optical biosensors have been released on the market, the most of them are large and expensive equipment. The future investigation will focus on achieving complete sensor and detection integration on a single chip to create mechanically stable, reproducible, and user-friendly devices. As a result of extensive research, integrated, portable biosensors will surely become frequent in our daily life and have a positive influence on how we live in the near future.

## Figures and Tables

**Figure 1 micromachines-14-00295-f001:**
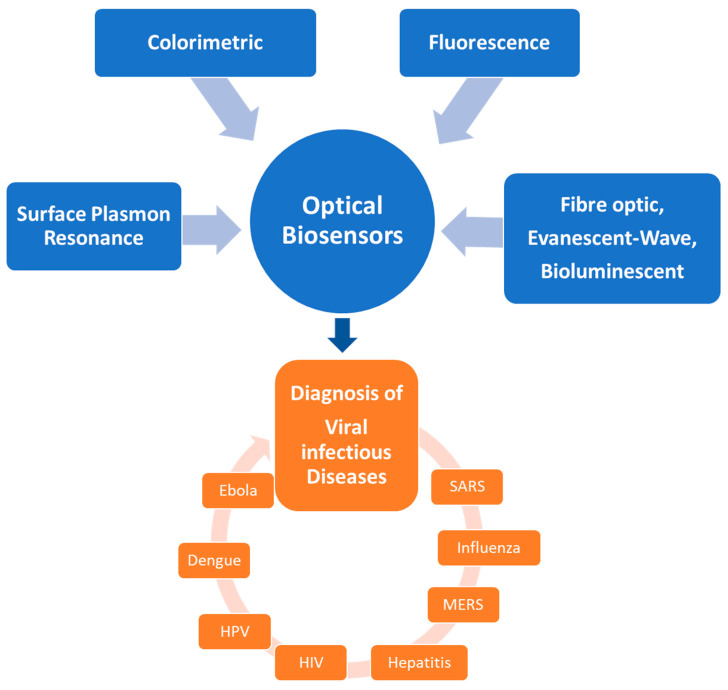
The optical biosensors developed for diagnostics of infectious viral diseases.

**Table 1 micromachines-14-00295-t001:** List of the colorimetric biosensors for virus detection.

Material	Virus/Disease	Recognition Element	Dynamic Range	LOD	Real Sample	Reference
AuNPs	Dengue	PNA	0–12 µM	0.12 µM	-	[57]
AgNPs	MERS-CoV, HPV	PNA	20–1000 nM (MERS-CoV), 20–2500 nM (HPV)	1.53 nM (MERS-CoV),	-	[59]
AuNPs	HPV genotype 16 and 18	DNA probe	10^5^ to 10^0^ copies	10^2^ (HPV 16) and 10^1^ (HPV 18)	clinical cervical tissue specimens	[62]
AuNRs	HBV	DNA	0.045 to 6.0 nM	15 pM	Human blood	[68]
AuNPs	HBV, HCV	DNA	-	HBV: 3.6 × 10^4^ fM; HCV: 3.6 × 10^5^ fM	Serum	[69]
AuNPs	Influenza	aptamer	-	2 × 10^6^ copies/mL	-	[70]
	Zika virus	Nucleic acids	-	5 PFU/mL	Saliva	[71]
AuNPs, AgNPs	Herpesvirus	DNA	10 pM–1 µM	1 nM	-	[72]
	Influenza A and B virus	aptamer	0.1 pg/mL– 1 mg/mL	0.30 pg/mL, 0.16 pg/mL	Patient samples	[73]
Copper nanoclusters	HBV	DNA	12 × 10^9^ to 12 × 10^13^ molecules	12 × 10^9^ molecules	Human serum	[74]
AuNPs	HCV	oligonucleotides	7.50 × 10^2^ to 2.00 × 10^6^ IU/mL	100 IU/mL (0.4 IU/µL)	Human blood	[75]
AuNPs	Cyprinid herpesvirus-3 (CyHV-3)	DNA	100 ng/µL to 0.1 fg/µL	10 fg CyHV-3-AuNP DNA (30 virions)	Infected fish tissues	[76]

**Table 2 micromachines-14-00295-t002:** List of the fluorescent biosensors for virus detection.

Material	Virus/Disease	Recognition Element	Dynamic Range	LOD	Real Sample	Reference
AuNRs	HBV	ssDNA	0.045 to 6.0 nM	15 pM	Human blood	[68]
Ag@SiO_2_ nanoparticles	H5N1 influenza virus	aptamer	2 to 200 ng/mL (in buffer) 3.5 to 100 ng/mL (in serum)	2 ng/mL (in buffer) 3.5 ng/mL (in serum)	Human serum	[83]
GO	Ebola virus gene influenza	dsDNA	30 fM–3 nM	1.4 pM	Human serum	[90]
AuNPs-GO	HIV-1 gene	dsDNA	50 fM–1 nM	15 fM	Human serum	[91]
AgNCs/GO	HIV	dsDNA	10–200 nM	1.18 nM	Human serum	[92]
g-C_3_N_4_ Graphitic carbon nitride (gC_3_N_4_) nanosheets	HBV gene	DNA	2–100 nM	1 nM	Human serum	[93]
rGONs	HCV	dsDNA	-	10 fM	Human serum	[94]
GO	Influenza virus H3N2 Henagglutinin gene	RNA	37–9400 pg	3.8 pg	-	[95]
QDs	HIV, HBV	DNA	10^3^ to 10^9^ copies/mL	1 × 10^3^ copies/mL	Human serum	[96]
QDs	Influenza (H1N1)	aptamer	10 nM–100 nM	3.45 nM	-	[97]
AuNPs	SARS-CoV-2	DNA	0.01–1 pM	160 fM	-	[98]

**Table 3 micromachines-14-00295-t003:** List of the Surface-Enhanced Raman Spectroscopy (SERS), Surface Plasmon Resonance (SPR)-based biosensors for virus detection.

Technique	Material	Virus/Disease	Recognition Element	Dynamic Range	LOD	Real Sample	Reference
SERS	Au-Gr film	Influenza	Hairpin DNA probe	0–60 amole	2.67 amole	-	[108]
SERS	-	Influenza	aptamer	-	10^4^ virus particles per sample	-	[109]
SERS	AuNPs	HBV	oligonucleotide	0.01 fM–6 µM	0.14 fM	serum	[110]
SPR	-	Epstein-Barr virus (LMP1 DNA)	LMP1 DNA probe	-	4.1 × 10^−5^ RIU	-	[111]
SPR	Au-Al bilayer	Dengue virus (NS1 protein)	DNA	0.1–10 µg/mL	-	Bovine whole blood	[112]
SPR	AgNPs	HIV	DNA	0.3–2.0 nM	195 pM	-	[113]
SPR	-	Avian influenza H5N1	aptamer	0.128 to 1.28 HAU	0.128 HAU	poultry swab samples	[114]
SPR	-	African swine fever (ASF) virus vp72 gene	ssDNA/LNA probe	0–15,000 copies/μL	178 copies/µL	Pig blood	[115]

## Data Availability

No new data were created or analyzed in this study.

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
