# Peer review of "Recent Progress on Optical Biosensors Developed for Nucleic Acid Detection Related to Infectious Viral Diseases"

_micromachines, 2023, doi:10.3390/mi14020295_

Round 1

Reviewer 1 Report

This review paper by Eksin and Erdem provides an interesting and very well-structured overview of current optical techniques for detecting nucleic acids, and places this discussion in the specific context of infectious viruses. The work is written at a technical level that I feel will be understandable for scientists from a broad range of backgrounds, ranging from biochemistry to optics, all without compromising on important scientific details.

My opinion is that the manuscript should be accepted for publication after a few minor revisions to the content, mostly to correct one or two vague statements and provide some additional physical context. I have detailed these below. I also feel the manuscript would benefit from a more thorough proof-reading to correct grammar mistakes (mostly missing or incorrect articles) but generally the work is understandable and I have only commented on issues below where they have affected my understanding.

On line 200, the term "high number of immigrants" is a bit vague. The correct word is probably "migrants", and the authors should clarify by what metric the number is "high". Is the number increasing or decreasing, and can a reference be provided?

On line 268, the authors refer to "colored signals from red to blue". The authors should clarify here that this is due to absorption by the NPs and is not emission of red or blue light, which an unfamiliar reader might assume.

On line 319, when discussing the paper by Zhou et al. It's not clear to me what is meant here without reading the referenced article; I think there's a word missing, and it should read "a silver shell"

In Table 1 - "Cupper" should read "Copper". There are missing units for the values of dynamic range for this example.

Section 3.4 - This section mentions some interesting and varied work, but I feel it should start with a clarification on how SPR-based sensing works. How does the binding of an analyte result in a colour change? The authors should explain the physical principle, with reference to refractive indices etc.

My apologies for correcting minor spelling mistakes in references, but these are easy to miss during proofreading (as I have learnt to my own cost!)

ref 86 - spelling of "amplification" and "fluorescence" incorrect

ref 87 - spelling error in "specific" twice

refs 103 and 104 - spaces are missing; "biosensor composed" and "highly accurate"

Author Response

Journal: Micromachines (ISSN 2072-666X)

Manuscript ID: micromachines-2120097

Type: Review

Title: Recent progress on Optical Biosensors developed for Nucleic Acid detection related to Infectious Viral Diseases

Authors: Ece Eksin, Arzum Erdem *

16th Jan 2023

The list of answers to the comments of reviewers

Thank you for valuable comments of Reviewer#1 and Reviewer#2. Manuscript is revised/corrected according to each comment pointed by reviewers. The revised/corrected parts in the manuscript are shown highlighted in yellow.

Reviewers' comments and suggestions:

Reviewer #1:

This review paper by Eksin and Erdem provides an interesting and very well-structured overview of current optical techniques for detecting nucleic acids, and places this discussion in the specific context of infectious viruses. The work is written at a technical level that I feel will be understandable for scientists from a broad range of backgrounds, ranging from biochemistry to optics, all without compromising on important scientific details. 

My opinion is that the manuscript should be accepted for publication after a few minor revisions to the content, mostly to correct one or two vague statements and provide some additional physical context. I have detailed these below. I also feel the manuscript would benefit from a more thorough proof-reading to correct grammar mistakes (mostly missing or incorrect articles) but generally the work is understandable and I have only commented on issues below where they have affected my understanding.

Answer: We would like to thank to the Reviewer #1 for their valuable consideration.

On line 200, the term "high number of immigrants" is a bit vague. The correct word is probably "migrants", and the authors should clarify by what metric the number is "high". Is the number increasing or decreasing, and can a reference be provided?

 Answer: We would like to thank to the Reviewer #1 for their valuable comments and suggestions. Accordingto yor suggestion, “immigrants” replaced with the word “migrants”. With the “high number of immigrants” statement, it was aimed to mention the large population of migrants. The reference 34 (34.Hasan, S.; Ahmad, S.A.; Masood, R.; Saeed, S. Ebola virus: A global public health menace: A narrative review. J Family Med Prim Care. 2019, 8, 2189-2201.) was cited as a supporting reference of our statement. In order to avoid misunderstanding the statement is revised as follows,

“Due to the large population of migrants, the illness has become a global public health threat.”

On line 268, the authors refer to "colored signals from red to blue". The authors should clarify here that this is due to absorption by the NPs and is not emission of red or blue light, which an unfamiliar reader might assume.

 Answer: According to your suggestion, more information about the colored signal by means of AuNPs is added as follows.

“Gold nanoparticles (AuNPs) are frequently used for fabrication of colorimetric biosensors using aggregation (or deaggregation) of the nanoparticles induced by formation of covalent or non-covalent bonds with the target substance. The aggregation of gold nanoparticles will change the color of the solution from wine red to blue, which corresponds with surface plasmon band changes from 523 nm towards 610–670 nm [50]. Therefore, the change at color from red to blue can be obtained as a convenient signal readout using AuNPs.”

On line 319, when discussing the paper by Zhou et al. It's not clear to me what is meant here without reading the referenced article; I think there's a word missing, and it should read "a silver shell"

Answer: According to your suggestion and to make it easier for understanding of readers, more details about the work reported by Zhou et al. have been provided. Revised version of the text is given below.

“Alkaline phosphatase (ALP) detected using a colorimetric test approach based on enzyme-induced metallization by Zhou et al. [63]. This method combined with immunomagnetic separation and further used for sensitive detection of avian influenza virus. The main principle of the developed technique was that the enzyme could induce a silver shell to be deposited onto the surface of AuNPs, which could cause a significant color change. This phenomenon helped the biometallization process in which ALP catalyzed silver deposition on the surface of AuNP seeds, and AgNPs exhibit greater extinction coefficient than AuNPs. When a longer incubation period or higher ALP concentration was used, the detection solution's color would change to brown or even black, which contributed to the further growth of the silver shell.”

In Table 1 - "Cupper" should read "Copper". There are missing units for the values of dynamic range for this example.

Answer: The typo is removed and the unit for the dynamic range is added.  

Section 3.4 - This section mentions some interesting and varied work, but I feel it should start with a clarification on how SPR-based sensing works. How does the binding of an analyte result in a colour change? The authors should explain the physical principle, with reference to refractive indices etc.

Answer: According to your suggestion the required addition related to the SPR technique is added. Newly added part is given below.

“The SPR technology has been in use for four decades. In 1983, SPR was initially employed to build an SPR-based sensor to detect biomolecular interactions [102]. Multiple manufacturers currently produce SPR instruments, and the SPR-based biosensor is the main optical biosensing technique. The SPR phenomenon occurs on the surface of metal (or other conducting materials) at the interface of two media (typically glass and liquid) in case of illumination by polarized light at a precise angle. This produces surface plasmons, which reduce the intensity of reflected light at a specific angle known as the resonance angle. This effect is proportional to the mass on the surface. A sensorgram can be created by plotting the change in reflectance, angle, or wavelengths versus time. The SPR technique provides direct, label-free, and real-time changes in refractive index at the sensor surface that is proportional to biomolecule concentration in all configurations. A practical SPR instrument consists of an optical detector that measures intensity shift, a sensor chip with a gold surface and a layer enabling ligand immobilization, and a fluidics system that allows for flow-through operation [102].”

My apologies for correcting minor spelling mistakes in references, but these are easy to miss during proofreading (as I have learnt to my own cost!)

 ref 86 - spelling of "amplification" and "fluorescence" incorrect

ref 87 - spelling error in "specific" twice

refs 103 and 104 - spaces are missing; "biosensor composed" and "highly accurate"

Answer: Thank you for your great attention. According to your suggestion all references are checked, and all typos are removed.

Reviewer 2 Report

This work contains an interesting matter. In order to be published, some modifications are required.

1.      Line 25: CT -> X-ray CT is better.

2.      Line 44: I do not understand miRNA. How about mRNA, tRNA, of cDNA?

3.      Line 75: There is meaningless space between “of” and “infection”?

4.      Line 407: There is no definition of SERS in the text.

5.      Table 3: Clarify the term of SPR, fiber optic, evanescent-wave, bioluminescent, and FOBS. There is only SPR in Table 3. How about fiber optic, evanescent-wave, bioluminescent, and FOBS?  

Author Response

Journal: Micromachines (ISSN 2072-666X)

Manuscript ID: micromachines-2120097

Type: Review

Title: Recent progress on Optical Biosensors developed for Nucleic Acid detection related to Infectious Viral Diseases

Authors: Ece Eksin, Arzum Erdem *

16th Jan 2023

The list of answers to the comments of reviewers

Thank you for valuable comments of Reviewer#1 and Reviewer#2. Manuscript is revised/corrected according to each comment pointed by reviewers. The revised/corrected parts in the manuscript are shown highlighted in yellow.

Reviewers' comments and suggestions:

Reviewer #2:

This work contains an interesting matter. In order to be published, some modifications are required.

 Answer: We would like to thank to the Reviewer #2 for their valuable consideration.

  1. Line 25: CT -> X-ray CT is better.

Answer: The required correction is done as follows:

“The X-ray computed tomography (X-ray CT),…”

  1. Line 44: I do not understand miRNA. How about mRNA, tRNA, of cDNA?

Answer: According to your suggestion, more example is given as below.

“In comparison to conventional bioassays, nucleic acid-based biosensors and biochips provide various advantages to genetic testing, including quick analysis times, low liquid handling, smaller sample volume requirements, and multiple detection options for specific nucleotide sequence of DNA or RNA (i.e., cDNA, miRNA, mRNA, tRNA), circulating tumor DNA, single nucleotide polymorphisms (SNPs), damaged DNA, etc. [6-8].”

  1. Line 75: There is meaningless space between “of” and “infection”?

Answer: According to your comment, the space is removed.

  1. Line 407: There is no definition of SERS in the text.

Answer: According to your comment, definition of SERS is added to the section 3.3 as follows:

“Surface-Enhanced Raman Spectroscopy (SERS) has been used to detect a variety of viruses, including the influenza, Adeno, West Nile, and rift valley fever viruses.”

  1. Table 3: Clarify the term of SPR, fiber optic, evanescent-wave, bioluminescent, and FOBS. There is only SPR in Table 3. How about fiber optic, evanescent-wave, bioluminescent, and FOBS?  

Answer: According to your suggestion the terms of SPR, fiber optic, evanescent-wave, bioluminescent, and FOBS are described in detail. The principles and more information about these techniques are added. The revised section is given below.

“Fiber Optic Biosensors (FOBS) detect the target (bio)molecule by using optical transduction techniques using optical fibers as the transduction component. The notion of total internal reflection (TIR) is employed in fiber optics to correlate the light intensity measured at the detector with the initial target concentration [116]. Fiber-optic biosensors are based on the measurement of absorbance, RI, fluorescence, chemiluminescence etc. and can be used for different biomolecule detection. With recent discoveries in the field of nanotechnology, the development of optical fibers with micro- or nano-sized dimensions has opened up a new route for detecting microorganisms, cells, in vivo biosensing [117].

In evanescent wave biosensors, specific ligands are immobilized to the sensor surface, and a solution of receptor is injected over the top. Subsequently, the binding is measured by recording changes in the refractive index, caused by the molecules interacting near the sensor surface within the evanescent field [118]. Evanescent wave fluorescence biosensors have evolved into a wide variety of devices since their discovery and initial application in the middle of the 1970s. Taitt et al. provided a comprehensive study of the evanescent wave fluorescence biosensors' characteristics, operation, and applications [119]. Evanescent wave FOBS are rely on evanescent wave detection methods. Total internal reflection at the exposed surface is how electromagnetic waves move within an optical fiber. The guided field in the core and the exponentially decreasing evanescent field in the cladding constitute the two components of light that propagate through an optical fiber. The cladding of a fiber is diminished or removed in evanescent wave FOBS, allowing the evanescent wave to interact with the surroundings. Thus, evanescent wave FOBS can rapidly and accurately detect these target analytes from complicated matrix samples, significantly improving the sensitivity, selectivity, and speed of the detection process.

Luminescience from fluorescent proteins or luminescent enzymes is commonly used in biological research for monitoring changes in the surroundings of cells. A simple method is to directly measure the luminescence from the sample solution in a test tube using a single photon detector by simple coupling optics. The luminescence detection with the optical-fiber based system enables to fully use the merits of compactness, high quantum efficiency, and low noise of avalanche photo diodes detectors [120]. Recombinant bioluminescent cells are used in the bioluminescent optical fiber technology, and an optical fiber is used to transmit the bioluminescent signal from the analyte. The detection of a particular molecule may then be performed by accurately detecting the emitted photons with a photon detector. In order to obtain a high sensitivity to the specific molecules or ions, a highly effective luminescence detection is required. The capability of detection in a low concentration of sample solution can be upgraded by an increase in sensitivity.”

Additionally, Table 3 is revised. New version of Table 3 consists of only SPR and SERS based biosensors, due to the lack of nucleic acid based Evanescent-Wave Optical Biosensors, Fiber optic biosensors, Bioluminescent optical fibre biosensors for infectious viral detection in the literature. However, these techniques are widely used in the field of diagnostics of infectious viral diseases. Therefore, it was aimed to mention at least principle of these techniques as optical biosensing techniques.
